# Development and Evaluation of an Enzyme-Linked Immunosorbent Assay Targeting Rabies-Specific IgM and IgG in Human Sera

**DOI:** 10.3390/v15040874

**Published:** 2023-03-29

**Authors:** Michelle D. Zajac, Maria Teresa Ortega, Susan M. Moore

**Affiliations:** 1Department of Diagnostic Medicine/Pathobiology, College of Veterinary Medicine, Kansas State University, Manhattan, KS 66506, USA; amthauer@vet.k-state.edu (M.D.Z.); mto5535@vet.k-state.edu (M.T.O.); 2Veterinary Medical Diagnostic Laboratory, College of Veterinary Medicine, University of Missouri, Columbia, MO 65211, USA

**Keywords:** rabies, ELISA, serum neutralization, immunoglobulin isotype, antibody kinetics

## Abstract

Immunity from rabies depends on rabies virus neutralizing antibodies (RVNA) induced after immunization; however, the influence of antibody isotype switching has not been extensively investigated. This has become particularly relevant with changes in World Health Organization (WHO) recommended rabies vaccine regimens that may influence RVNA isotype kinetics, potentially affecting the peak, and longevity, of RVNA immunoglobulin (IgG) levels. We developed rapid and reliable assays for quantifying the anti-rabies IgM/IgG class switch in human serum based on an indirect ELISA technique. The immune response was tracked in ten individuals naïve to the rabies vaccine by quantifying serum titers weekly, from day seven to day 42 post-immunization, using a serum neutralization assay and the ELISA IgM/IgG assays. The average RVNA IU/mL levels were at D0 ≤ 0.1, D7 0.24, D14 8.36, D21 12.84, D28 25.74 and D42 28.68. Levels of specific IgM antibodies to rabies glycoprotein (EU/mL) were higher, on average, at D7, 1.37, and from D14, 5.49, to D21, 6.59. In contrast, average IgG antibodies (EU/mL) predominated from D28, 10.03, to D42, 14.45. We conclude that levels of anti-rabies IgM/IgG at D28 characterize the isotype class switch. These assays, combined with serum neutralization assays, distinguished the RVNA levels in terms of the IgM/IgG responses and are expected to add to the diagnostic repertoire, provide additional information in establishing rabies vaccine regimens, both post- and pre-exposure prophylaxis, and contribute to research efforts.

## 1. Introduction

Rabies infection is nearly 100% fatal once symptoms appear. Post-exposure prophylaxis (PEP), including a rabies vaccination series and rabies immunoglobulin administration (in those with no previous rabies vaccination), has been shown to be nearly 100% effective in preventing disease. Eliminating human rabies deaths caused by canine rabies, as targeted by the Zero by 30 effort launched by WHO, the World Organisation for Animal Health (WOAH) and the Food and Agriculture Organization (FAO) in 2015, hinges upon successful vaccination and availability of reliable, readily accessible diagnostics to detect and monitor adequate levels of protection in vaccinated individuals and animals [1,2]. Canine vaccination has been proven to be the most effective method to control the disease at source in endemic countries. In humans, rabies prevention requires pre-exposure prophylaxis (PrEP) (vaccination) for those at high risk of rabies exposure and PEP (vaccination and administration of rabies immunoglobulin) for those who have potentially been exposed [3,4].

Rabies PrEP vaccination regimes have varied over time. Purified cell-culture and embryonated egg-based rabies vaccines currently in use have been shown (using the 3-dose regimen) to confer both long-lived immunity and efficacy, and have been proven to be safe for usage in a large range of age groups [5,6,7]. The two-dose regimen has produced RVNA longevity through robust booster response up to 3 years post-PrEP [4,8]. Rabies vaccines can be administered by either an intramuscular (IM) or intradermal (ID) route. For IM, it has been established that a dose of >2.5 IU/mL requires a full vial (from 0.5 mL to 1 mL), whereas only 0.1 mL is required via the ID route to confer the same level of protection [9,10]. The World Health Organization and the Advisory Committee on Immunization Practices (ACIP) recommend the administration of a two-site vaccination on day 0 and day 7 (IM or ID) [3,4].

PrEP is administered for those persons deemed a high risk of exposure, such as laboratory workers and those traveling to rabies endemic areas [11]. Studies conducted in the United Kingdom analyzed rabies antibody titers from humans and have concluded that as early as day four post vaccination anti-rabies IgM isotype is detectable and anti-rabies IgG isotype was identified as early as day seven [12]. IgG is the most effective at neutralizing the virus and also circulates the longest in serum whereas IgM is unable to penetrate into tissues and prevent rabies infection. Since RABV does not disseminate via the bloodstream and the IgM remains in the vascular circulation, the short lived IgM isotype’s role in viral inactivation and elimination could be more effective in cases in which vaccine-induced immunity to rabies virus (RABV) relies on a rapid short-term response for protection [11,12,13]. In inactivated RABV immunization the response is T cell-dependent; antigen-specific B cells get activated by primed T-cells at the borders of secondary lymphoid organs, then they recirculate as both early plasma cells and activated B cells. Activated B lymphocytes in turn activate T lymphocytes (CD4+) and T helper type 2 (Th2) via the MHC II (major histocompatibility complex II) [14,15]. Class switching from IgM to IgG occurs in the immunoglobulin-secreting B cells via molecules produced in the microenvironment. For rabies, IgM is generally considered a small player in the protective immune response, therefore the target is modifying the vaccination schedule to drive the response to sustained IgG production [16,17]. The preliminary IgM response does correlate with the early rise in viral titer as measured by the gold standard RFFIT (cell-based), which detects both IgG and IgM [18]. Studies by Dorfmeier, et al., have shown the potential practicality of an IgM-based rabies vaccine in post-exposure prophylaxis where a rapid immune response is key to protection against exposure to the virus [16], suggesting additional knowledge of the kinetics of the antibody response could be useful.

Vaccine effectiveness is challenged within the population by the individual’s unique immunological response to vaccination. The different factors that affect the individual populations are due to both intrinsic host factors and extrinsic vaccine factors [19]. Some intrinsic host factors and medical conditions, such as HLA type, pregnancy, diabetes type 1, diabetes type 2, chronic infection, cancer and having a compromised or impaired immune response, contribute to the efficacy of vaccination.

The ACIP updated the regimen for PrEP in 2022, from a three-dose series to a two-dose series, similar to the current WHO PrEP recommendations. The shortened vaccine regimens may influence peak levels of rabies virus neutralizing antibody (RVNA) and Ig class kinetics. Previous studies have indicated that peak levels of antibody-producing plasma cells are achieved after two vaccinations; however, memory B cell frequency increases after a third vaccination [20]. Information about when RVNA isotype switching occurs can provide information on the timing and number of vaccinations needed to elicit sufficient recall in the immune response months/years after PrEP, the rationale for which requires a long-term response. To date, there has not been a rapid and reliable diagnostic assay to detect and quantify RVNA isotype switching kinetics from IgM, the primary antibody response, to IgG that occurs during the rabies vaccination series. Rabies antibody IgM to IgG class switch is relevant because it provides knowledge of rabies virus vaccination immune responses and aids understanding of immunization responses using currently approved vaccines and will guide future research.

## 2. Materials and Methods

### 2.1. Serum Samples

Serum samples used in the study were obtained from ten subjects vaccinated according to the ACIP regimen for pre-exposure (on days 0, 7, 21, or 28, as available) as per the ACIP guidelines effective at the time [4,21]. Blood was collected on days 0, 7, 14, 21, 28 and 35 (in some instances a day 42 sample was also obtained) post-initial (day 0) vaccination. A pool of serum from unvaccinated subjects was used as the rabies antibody-negative control. Samples were de-identified as per the protocol reviewed and approved by Kansas State University IRB#9819. Samples were heat-inactivated for 30 min at 56 °C to remove complement factors, which have been proven to interfere with serum neutralization assays.

### 2.2. RFFIT Serological Testing

RFFIT, using CVS-11 as the challenge virus strain, was used to test all serum samples for the baseline rabies virus neutralizing antibody (RVNA) titer value, as previously described [22]. The RFFIT assay has been validated in the Kansas State University Rabies Laboratory. Briefly, 100 μL of each serum sample was diluted in serial five-fold dilutions in 96-well microplates and 100 µL of each serum dilution was loaded into 8-well Lab-Tek chamber slides (Nunc Lab-Tek Chamber Slide System, catalog# 177445), followed by addition of 100 μL of the challenge virus (ATCC, product code VR 959), at a concentration of 50 TCID_50_. The resulting serial dilution represents serum dilutions of 1:5, 1:25, 1:125 and 1:625. A 90 min incubation period to allow for virus neutralization by any RVNA in the samples was followed by the addition of 200 μL DEAE-treated BHK-21 [C-13] cells (ATCC, product code CCL-10 (5 × 105/mL) followed by incubation for 24 h. Detection of the residual virus under fluorescence microscopy was achieved by staining the acetone-fixed slides with FITC-conjugated anti-N rabies antibody (Millipore, Burlington, MA, USA, product code 5500). Counts of virus-detected fields were transformed to IU/mL results using the Reed and Muench formula for endpoint titer, and titer results were compared to an international RVNA standard sample titer tested alongside the samples. High-titer sera were pre-diluted in cell media to obtain a readable result within the linear range of the assay (0.1 IU/mL to 15.0 IU/mL) as defined per method validation.

### 2.3. Indirect ELISA Serological Testing

The Platelia^T^ Rabies II Kit (Marnes-la-Coquette, France, 3551180) is an indirect ELISA using rabies glycoprotein as the antigen coating the wells and a Protein A-peroxidase conjugate, referred to below as Protein A ELISA. It is used for in vitro detection and titration of anti-rabies virus glycoprotein IgG in human serum and plasma and was used following the manufacturer’s instructions as previously described [23]. The optical density (OD) readings were measured with a Bio-Tek ELx808 microplate reader (Winooski, VT, USA). The EU/mL values were calculated per the kit instructions and the supplied calculation spreadsheet, which involved a comparison of the sample optical density (OD) reading against a standard curve of positive standards supplied in the kit (range: 0.125 to 4 EU/mL). Serum samples with initial test results above 4 EU/mL were pre-diluted in the kit diluent to obtain a readable result within the linear range of the assay. The Protein A ELISA kit has been validated in the Kansas State University Rabies Laboratory.

### 2.4. Development of Anti-Rabies IgM and IgG ELISA for Human Samples

The Protein A ELISA was modified for the detection of rabies-specific human IgM and IgG antibodies. Indirect ELISA secondary (detection) antibodies consisting of anti-human IgG (Sigma, St. Louis, MO, USA, product code A0170) and anti-human IgM antibodies (Sigma, product code A6907), conjugated to horseradish peroxidase (HRP), were used to detect IgG and IgM, respectively, replacing the kit’s peroxidase-Protein A. Internal standards at known rabies antibody levels and a negative control were included on each ELISA plate to assess inter-assay precision. Sample dilutions were prepared for high-titer sera to bring the antibody level within the assay range (0.625 to 20 EU/mL for IgG and 0.250 EU/mL to 8.0 EU/mL for IgM). The non-linear curve fit standard curve in Gen5 software was selected for analysis of OD readings (from the available Gen5 standard curve analysis types) based on the tightest fit to the Bio-Rad proprietary standard curve analysis method (Bio-Rad Platelia Kit supplied spreadsheet, Marnes-la-Coquette, France) [24]. Samples with initial test results above the identified upper range of the assays were pre-diluted in the Protein A ELISA kit diluent to obtain a readable result within the linear range of the assay [24].

#### 2.4.1. Rabies Anti-IgG ELISA Development

Subject serum samples drawn at day 21 or later were screened in the Protein A ELISA to select samples with high EU/mL results to serve as anti-rabies human IgG antibody controls along with the Protein A ELISA kit control sample (4.0 EU/mL). In addition, an anti-rabies IgG internal control sample (RIgG) with a Protein A ELISA value of 20 EU/mL was serially diluted to prepare a standard curve. Serial two-fold dilutions (1:1600 to 1:100,000) of anti-human IgG-HRP conjugate were used to identify the IgG-HRP conjugate dilution required to produce comparable results against the Protein A ELISA. A sample with no rabies antibody (negative control) was included to demonstrate specificity of the conjugate to the rabies antibodies. Tests for linearity, accuracy and precision were performed. Gen5 was used to construct the standard curve and for interpolation of sample data.

#### 2.4.2. Rabies Anti-IgM ELISA Development

Subject samples drawn on days 7, 14 and 21 post-vaccination were screened by RFFIT testing to identify samples with high RVNA titers, that is samples with RVNA IgM and/or IgG. Then, the samples were screened by Protein A ELISA to identify samples with low Protein A ELISA titers, that is samples with low IgG binding antibody. Screened serum samples served as the assay positive controls for ELISA development. A checkerboard experiment was used to optimize anti-human IgM-HRP conjugate assay dilution. Samples and anti-IgM-HRP conjugate were used in two-fold serial dilutions (1:2 to 1:64) against anti-IgM conjugate serial two-fold dilutions to identify the target conjugate dilution, followed by optimization experiments to develop a standard curve. A sample with no rabies antibody (negative control) was tested to ensure anti IgM-HRP conjugate specificity; no background reactivity was noted with the conjugate dilutions evaluated. Tests for linearity, accuracy and precision were performed. Gen5 was used to construct the standard curve and for interpolation of sample data.

### 2.5. Statistical Methods

GraphPad Prism, version 9.5.0, and Gen5 software were used for the construction and evaluation of standard curves. Coefficient of variation (%) was calculated to describe the level of variability within the measured isotype titers and to determine the assay precision. CV% was calculated by the formula standard deviation/average multiplied by 100, and % recovery (%R) by the formula actual EU/mL/expected EU/mL multiplied by 100.

## 3. Results

All subject serum samples (from days 0, 14, 21, 28 and 42) were tested using the RFFIT, the Protein A ELISA and the developed IgG and IgM ELISAs, see Appendix A for individual results and Appendix A for averages ±2 standard deviations per day post vaccination (DPV).

### 3.1. IgG Assay

The closest standard curve fit was obtained using the IgG-HRP conjugate dilution of 1:4800 against two-fold serial dilutions of the RIgG internal control sample (Figure 1). The RIgG internal control sample results obtained with the IgG ELISA and compared against the Protein A ELISA guided the identification of the curve fit. The measurable range was 0.625 to 20 EU/mL.

All subject serum samples (from days 0, 14, 21, 28 and 42) were tested in the developed IgG ELISA, see Appendix A for individual results and Appendix A for averages ±2 standard deviations per day post vaccination (DPV). Linearity and accuracy of the assay were determined by testing three IgG-positive samples: high (RAE-10), medium (RAE-27) and low (RAE-16) EU/mL values, in two-fold dilutions (neat to 1:128). For linearity, the EU/mL values were plotted, and the R^2^ values were calculated in Excel; the R^2^ values for the three samples (high, medium, and low EU/mL values) were 0.98, 0.94 and 0.97, respectively. For accuracy, the % recovery (%R) of the measured EU/mL against the expected value (the Protein A ELISA result) was found to be 85.77%, 113.12% and 52.94% for the high, medium and low IgG positive samples, respectively. The precision of the assay, determined by calculating the CV% of duplicate results of the three samples, ranged from 5.24% to 7.76%.

### 3.2. IgM Assay

Four of the 10 serum samples with the highest RFFIT and low Protein A ELISA kit results were selected as the IgM control samples. The IgM positive control samples were tested in duplicate in the IgM ELISA, using IgM-HRP conjugate in dilutions starting with IgM-HRP manufacturer’s recommended dilution of 1:10,000. The observation that higher concentrations (lower dilutions) of the IgM-HRP conjugate raised the OD reading to approximately 1.8 for the lowest dilution of the sample while maintaining good curve fit led to experimenting with lower IgM-HRP dilutions with two-fold serial dilutions of sample RAE-21 (estimated ~8.0 EU/mL based on the RFFIT result), see Figure 2. The IgM-HRP 1:1000 conjugate dilution provided a good curve fit and allowed the IgM positive controls to reach the optimal ∆ OD reading, near 4.0, to allow for the construction of a standard curve, and was selected for use in the IgM ELISA with a measurable range of 0.250 to 8.0 EU/mL.

All subject serum samples (from days 0, 14, 21, 28 and 42) were tested using the IgM ELISA, see Appendix A. Linearity and accuracy of the assay were determined by testing three IgM-positive samples: high (RAE-21), medium (RAE-4) and low (RAE-15) EU/mL values), in two-fold dilutions (neat to 1:128). For linearity, the EU/mL values were plotted, and the R^2^ values were calculated; the R^2^ values for the three samples’ results were 0.95, 0.99 and 0.96, respectively. The precision of the assay, determined by calculating the CV% of duplicate results of the three samples, ranged from 3.83% to 13.28%. Accuracy was not determined due to the unavailability of a validated rabies IgM reference assay.

### 3.3. IgM to IgG ELISA Ratio

The IgM to IgG ratio, using the mean EU/mL values of all subjects from each assay, displayed with the means (IU/mL) from the RFFIT assay (Figure 3), illustrates that IgM is the first Ig produced after rabies vaccination, as early as 7 days post-vaccination. IgM levels increased during the median sampling period, between day 7 and day 21 post-vaccination, for most subjects. Table 1 displays the individual IgM/IgG class switching variation between vaccine recipients, primarily between vaccine recipients at days 21 to 28 post-initial vaccination. The individual variability in the initial response to PrEP can be striking, as can be seen by comparing the three subjects’ rabies antibody kinetics, see Figure 4. Subject 1 has a higher RVNA response compared to the combined IgM and IgG response, whereas Subject 6 has a lower RVNA response, as would be expected from the IgM and IgG responses; Subject 3 also has an RVNA response that could be predicted from the IgM and IgG levels. See Appendix A for all the individual response graphs.
viruses-15-00874-t001_Table 1Table 1IgM/IgG EU/mL ratio (percent) per vaccine recipient per day post vaccination (DPV).
SubjectDPV123456789107
522.439.9221.4373.239.9

113.439.910

119.8






14–16421.1537.3191.7369.4
449.8337.339.9
25.221383.3278.4107.6226.371.7147.5

96.8
28–313584.820.921.434.2364.526.8134.1
49.14216.125.56.716.4
15.6

10.9
Gray shading indicates no sample available for that timepoint.

### 3.4. RVNA Isotype Switching Kinetics

IgM is the predominate class of human immunoglobulin present in the early stages of the vaccine-induced immune response, day 7–21 post-vaccination (Figure 3 and Table 1). The IgG response predominates at 28–42 days post-initial rabies virus vaccination. Class switching was evidenced between days 21 and 28. Thereafter the IgM to IgG ratio decreased as days post initial rabies vaccination progressed. The combined activity of IgM and IgG rabies-specific antibodies at days 0 to 21 post-initial vaccination contributed to the RVNA neutralizing ability as measured by RFFIT. The fully developed IgG response by day 21–28, see Figure 3, was associated with stronger rabies neutralization (RFFIT) and with lower quantities of IgG antibody compared to IgM, illustrating affinity maturation due to selection for and clonal expansion of these highly neutralizing IgG antibodies.

## 4. Discussion

It has long been understood that rabies antibodies, particularly neutralizing antibodies, are of primary importance in immunity to rabies virus infection [25,26,27]. Less studied is the effect of antibody class-switch kinetics in this protective role. Class switching from IgM to IgG occurs rapidly via immunoglobulin-secreting B cells, which alludes to the idea that IgM is a small player in the protective immune response due to vaccination [28]. The antibody response to PEP vaccination is characterized by a preliminary IgM response, and the rise in titer correlates in timing with an early rise in viral titer after rabies exposure and infection [16,17]. Dorfmeier et al. have shown the potential for the usefulness, in mice studies, of an IgM-based rabies vaccine in post-exposure prophylaxis where a rapid immune response is key to protection against exposure to the virus [29]. Characterizing the typical timing of the IgM and IgG responses to rabies vaccination can provide knowledge of the effects of changes in vaccine regimens, vaccine types, and PEP types (e.g., polyclonal versus monoclonal).

The rabies antibody kinetics identified in this study demonstrate similarity to published studies into the antibody response to human viral and bacterial infections. In rotavirus infection, IgM peaks in early collected serum samples and falls to low levels in the late infection-stage samples whereas IgG was present in the later days as detected by a combination of ELISA and serum neutralization methods, which is similar to the findings in our study for the response to rabies vaccination [30]. Antibody kinetics obtained via ELISA for the hepatitis E virus show the typical pattern of quick increase in IgM with seroconversion to IgG [31,32]. Enzyme immunoassays have been used to detect IgG against the Epstein–Barr virus and show the same trend of low IgG detected in newly infected patients and high IgG detection in persons with reactivation of the virus or previous exposure to the virus [33]. IgM and IgG are detected at the same time in both reactivation and after initial infection [33], which follows the same immunological trend for rabies upon initial infection or vaccination followed by booster dose(s). Despite a few mixed findings in the co-presence of IgM and IgG in early-stage infection *Treponema pallidum* infection leading to Syphilis, it is clear that IgM is only present in early infection, with IgG being the primary immunoglobulin detected in late infection [34,35,36]. ELISA (or a similar immunoassay) was the chosen method for each of the examples in determining the IgM versus IgG responses. This method is quantitative, gives an objective value from a calibrated reader, is less labor intensive than similar in vitro methods, and has been shown to have good specificity and sensitivity [23,37].

Purified cell-culture and embryonated egg-based rabies vaccines that are currently in use confer both long-lived immunity and efficacy, and have been proven to be safe for usage in a large range of age groups [6,38,39]. These can be administered by either an intramuscular (IM) or intradermal (ID) route [9,27]. To date, much of the knowledge of rabies vaccine efficacy has been obtained from PEP clinical trial studies. PrEP immunogenicity has come from studies on the initial response, and from clinical trials, but there is limited published data on the longevity of the antibody response. Research has shown that there is individual variability in the antibody response consisting of high, moderate, and low responders [40,41], the practical importance of which can only be extrapolated from animal studies [42,43,44].

By utilizing a variety of validation parameters for evaluating the use of the ELISA method in detecting rabies-specific IgG and IgM in human sera, method conditions were optimized to measure the concentrations of both IgG and IgM for each sample and accurately determine the antibody isotype kinetics of the immune response to rabies vaccination. Investigation of the immune reaction of each individual’s response to vaccination has contributed to further our understanding of the variability of immunoglobulin class switching between individuals and the specific timing of that class switch from, IgM to neutralizing IgG, via clonal expansion through affinity maturation in the lymphoid tissue for these best fit immune cells. A limitation of this study is the low sample size; however, these results are a starting point in being able to accurately predict high versus low responders’ effective immune defense and possibly identify kinetics of rabies antibody development related to longevity, to better classify an individual’s rabies protection status. This data will also help guide future research on preventative and treatment options as well as guide further development of anti-rabies immunological assays.

Future improvements, such as increased subject numbers, would aid in determining trends in non-conformant patient samples, such as is the case with samples RAE-55 through RAE-57 collected from a single patient all showing lower titration values across sampling days. Upon collection of the data from these subjects, further testing and analysis can be conducted to investigate other biologic players in the immune response, for example, cytokines, to expand our knowledge of atypical responses to rabies vaccination. This understanding would drive further research and development of more effective vaccines or vaccine schedules. In addition, further work to establish a qualified rabies IgM reference serum would improve standardization, and hence the accuracy, of rabies IgM assays. Validation of the IgM/IgG ELISA in a high-throughput automated ELISA system would also provide practical usefulness and increase the accessibility of these assays.

As demonstrated, the modified rabies enzyme-immunosorbent assays (ELISAs), with anti-IgM or anti-IgG secondary conjugates paired with the appropriate standard curve samples, are quantitative, obtain an objective value from a microplate reader and display similar specificity and sensitivity to the reference Protein A ELISA. Procedural steps and test conditions were optimized to detect, with confidence, the concentrations of both IgM and IgG for each sample and to accurately determine the kinetics of the immune response to rabies vaccination. Rabies antibody IgM to IgG class switch is relevant because it provides knowledge of the antibody response, particularly the isotype kinetics, to rabies vaccination. Understanding immunization responses to the different rabies vaccine regimens and vaccine types currently and in the future will aid in establishing efficient vaccine schedules and administration as the Zero by 30 effort progresses.

## Figures and Tables

**Figure 1 viruses-15-00874-f001:**
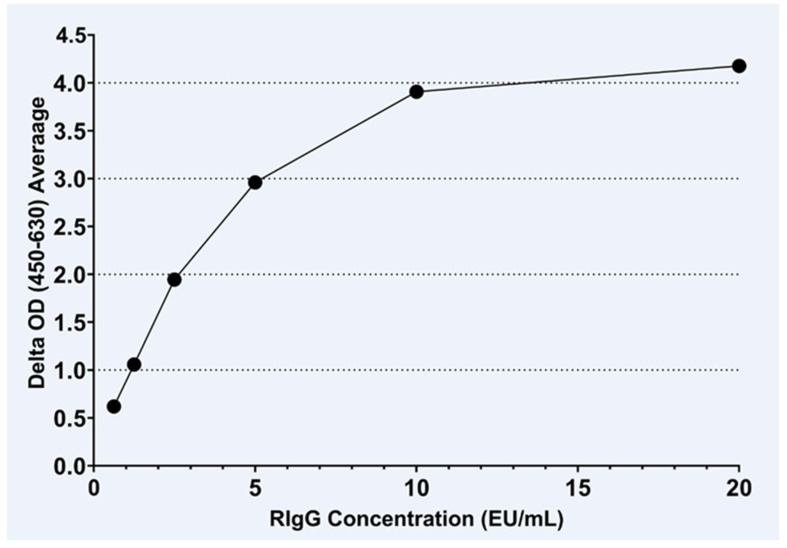
The IgG standard curve was constructed using the identified optimal dilution of the anti-human IgG-HRP conjugate to measure the OD value of serial two-fold dilutions of the RIgG internal standard (20 EU/mL per the Protein A ELISA); the EU/mL values were calculated by Gen5 software using non-linear curve fit standard curve analysis.

**Figure 2 viruses-15-00874-f002:**
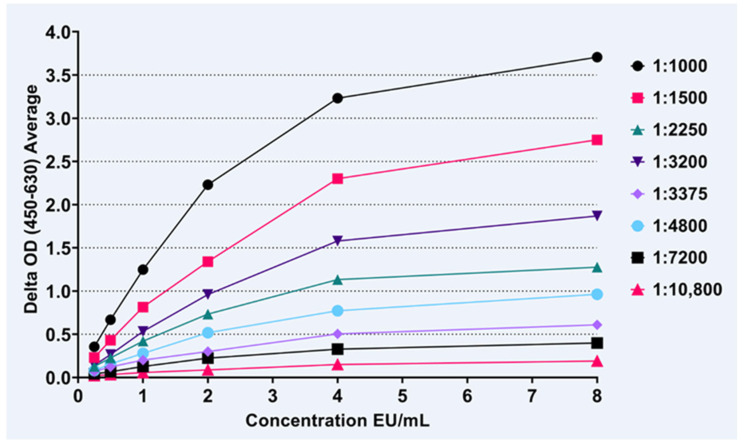
Anti-human IgM-HRP conjugate was tested, in dilutions ranging from 1:1000 to 1:10,800, against sample RAE-21 (estimated at ~8 EU/mL based on the RFFIT results) in serial two-fold dilutions, to identify the optimal concentration for the IgM ELISA.

**Figure 3 viruses-15-00874-f003:**
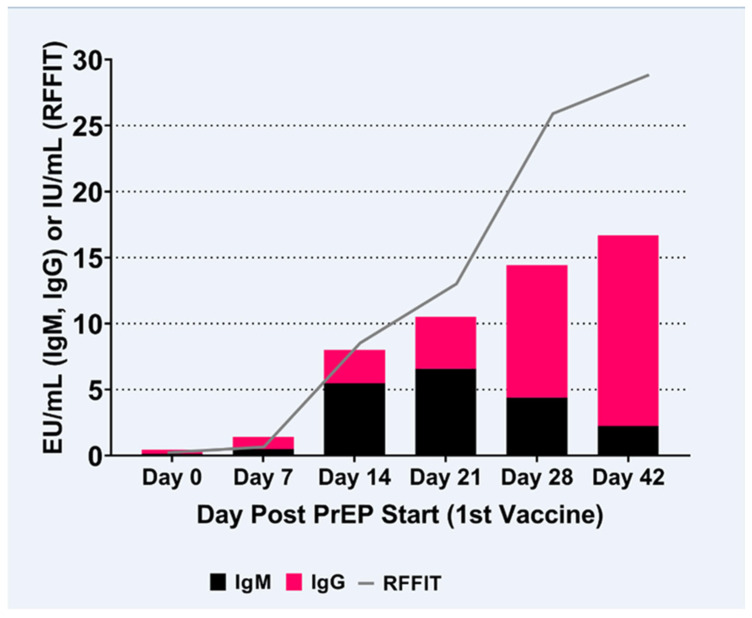
RVNA isotype switching kinetics displayed as average results per day post-vaccination (black bar = IgM, magenta bar = IgG, gray line = RFFIT). Note: Day 0 antibody activities in the assays are represented as ≤values (non-zero values).

**Figure 4 viruses-15-00874-f004:**
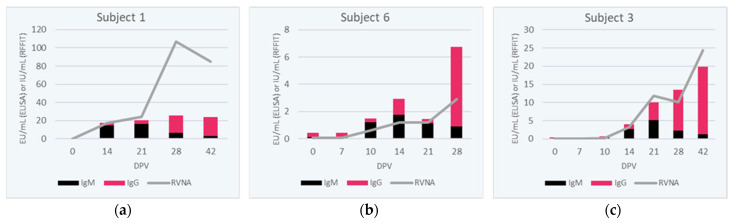
Individual variation in rabies antibody kinetics post-PrEP demonstrated in three subjects, 1 in panel (**a**), 6 in panel (**b**), and 3 in panel (**c**). The IgM ELISA result is in black, the IgG ELISA result is in magenta in the stacked bar, and the RVNA (RFFIT) result is the gray line.

## Data Availability

Data is contained within the article or Appendix A.

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
