# Peer review of "Development and Evaluation of an Enzyme-Linked Immunosorbent Assay Targeting Rabies-Specific IgM and IgG in Human Sera"

_viruses, 2023, doi:10.3390/v15040874_

Round 1

Reviewer 1 Report

The co-authors describe developing and using ELISA methods to detect the different antibody isotypes in the human immune response to rabies vaccine. I have no major concerns with the manuscript. One comment this manuscript would be more impactful if the co-authors used the method to compare PrEP regimens as they discuss in the manuscript. Minor editorial comments in the attached file.

Author Response

Response to Reviewer 1 Comments

Point 1: Abstract – proofreading comments to eliminate sentence at line 15 and remove parentheses at line 20-23 and rephrase portion of sentence at line 26.

Response 1: Thank you for the suggestions to improve the paper. All changes have been made.

Point 2: Introduction – proofreading comments at lines 31, 33, 51, 70 to 76, and 83 for rephrasing or corrections to text.

Response 2: Thank you for the suggestions to improve the paper. All changes have been made.

Point 3: Materials and Methods – proofreading comments at lines 115 (change if to of), 167 (rephrasing), 174 (removal of parentheses), and 210 and throughout (add superscript to R2.

Response 3: Thank you for the suggestions to improve the paper. All changes have been made.

Point 4: Figure 4 – Correct the legend regarding the color of bars related to the IgM and IgG values, as well as in the supplementary figure.

Response 4: Thank you for catching this error, corrections have been made.

Point 5: Discussion – proofreading comments at lines 316, and 319 to remove abbreviations as they are unnecessary. 

Response 5: Thank you for the suggestions to improve the paper. All changes have been made.

Reviewer 2 Report

Based on the clinical samples of 10 individuals, the neutralizing antibody and antiviral glycoprotein IgM and IgG antibody levels in human serum at different time after immunization were detected by commercial kit and improved ELISA based on commercial raw materials. This manuscript concluded that levels of anti-rabies IgM/IgG at D28 characterize the isotype class switch. Overall, the manuscript is well written, concise in its content, and shows compelling data, the work has some innovation. The tables and figures are clearly presented. However, I have minor comments which should be addressed.

1. Since the materials used in this manuscript are not homemade, but mostly commercial products, such as anti-IgM, IgG antibodies, and standard products used for quantitative purposes, it is necessary to clearly label the product's biomerchant or source.

2. Due to the lack of sample size at some time points, especially the 10th critical time point, the overall sample size was insufficient.

3. What impact do the findings of this study have on the decisions about rabies vaccine regimens?

Author Response

Response to Reviewer 2 Comments

Point 1: Since the materials used in this manuscript are not homemade, but mostly commercial products, such as anti-IgM, IgG antibodies, and standard products used for quantitative purposes, it is necessary to clearly label the product's biomerchant or source..

Response 1: Thank you for the suggestions to improve the paper. All supplier’s information have been added.

Point 2: Due to the lack of sample size at some time points, especially the 10th critical time point, the overall sample size was insufficient.

Response 2: Thank you for the suggestions to improve the paper. Additional comments in the discussion about the limitation of sample size were added.

Point 3: What impact do the findings of this study have on the decisions about rabies vaccine regimens?

Response 3: The kinetics of the IgM and IgG switch can provide information about the cellular response to rabies vaccine and potentially lead to clarification of vaccinees that will develop long-term response versus those with short-term response regarding circulating antibody. This may lead to individual vaccine regimens based on the response to assure long-term response for every vaccinee.

Reviewer 3 Report

The presented work describes a method to measure the level of anti-rabies G protein IgM and IgG in serum. The work is mostly devoted to developing an assay, but the details of the assay are not adequately described. The information about reagents (like antibody suppliers) is lacking. The chapter describing the optimized protocol, perhaps as a supplementary file, would greatly benefit the readers.

Author Response

Response to Reviewer 3 Comments

Point 1: The presented work describes a method to measure the level of anti-rabies G protein IgM and IgG in serum. The work is mostly devoted to developing an assay, but the details of the assay are not adequately described. The information about reagents (like antibody suppliers) is lacking. The chapter describing the optimized protocol, perhaps as a supplementary file, would greatly benefit the readers.

Response 1: Thank you for the suggestions to improve the paper. All supplier’s information have been added. Full details of the optimizing experiments can be found at http://hdl.handle.net/2097/40217 . A reference to the theses has been added.